# Head and body structure infants' visual experiences during mobile, naturalistic play

Chuan Luo, John M. Franchak 🔟 *

Department of Psychology, University of California, Riverside, Riverside, California, United States of America

* john.franchak@ucr.edu

## Abstract

Infants' visual experiences are important for learning, and may depend on how information is structured in the visual field. This study examined how objects are distributed in 12-month-old infants' field of view in a mobile play setting. Infants wore a mobile eye tracker that recorded their field of view and eye movements while they freely played with toys and a caregiver. We measured how centered and spread object locations were in infants' field of view, and investigated how infant posture, object looking, and object distance affected the centering and spread. We found that far toys were less centered in infants' field of view while infants were prone compared to when sitting or upright. Overall, toys became more centered in view and less spread in location when infants were looking at toys regardless of posture and toy distance. In sum, this study showed that infants' visual experiences are shaped by the physical relation between infants' bodies and the locations of objects in the world. However, infants are able to compensate for postural and environmental constraints by actively moving their head and eyes when choosing to look at an object.

## Introduction

Infants are immersed in a world full of objects to explore. Visual attention is one of the channels through which objects are selected for further processing while filtering out other potential distractions. Infants develop the ability to selectively attend to visual stimuli early in the first year of life [1, 2], which is important for cognitive development [3–5]. However, most of our knowledge about infant visual selection comes from studies using screen eye trackers (SETs). In a typical SET study setup, infants are seated in a chair or on a caregiver's lap while they look at images or videos on a screen. Although it is a useful technique to examine some aspects of infants' selective attention [6, 7], it precludes measuring the *gross motor* aspects of attention and how they shape infants' visual experiences of objects in daily life [8].

First, motor tasks shape where infants need to look. Screen-viewing studies do not place demands on the visual system for guiding gross motor action. In real-world adult studies of vision and action, visual attention is needed to guide the hand to objects that are relevant to an ongoing task [9–11]. When making a sandwich, people looked at bread before reaching for it and then fixated peanut butter jar while putting down the bread slice on the plate [12]. When building block models, adults moved their eyes in sequence with hand movements to pick up

**Data Availability Statement:** The original data from this study are available through the Databrary video data repository (DOI: 10.17910/B7.135), and the new coding files, aggregate data, and analysis

script from the current study are available on OSF (DOI: 10.17605/OSF.IO/3M8JB).

**Funding:** The author(s) received no specific funding for this work.

**Competing interests:** The authors have declared that no competing interests exist.

and position each block [13, 14]. Like adults, infants move their eyes to look at objects while reaching [15, 16]. Infants may selectively attend to objects to engage in object manipulation and explore object properties [15, 17–20]. Second, body posture (e.g., prone, sitting, and upright) shapes infants' visual attention [21–24]. Whereas in a screen-viewing study participants are seated with information easily viewed in front of the face, real-world visual attention is shaped by the orientation of the body. While prone on all fours, infants struggle to see faces because holding up the head to look at caregivers' faces is effortful [21, 25]. Looking at caregivers' faces is more frequent while sitting or upright when it is motorically easier to bring faces into view. Also, compared to walking infants, crawling infants look more often at the floor and have less visual access to distant and elevated objects [25].

Thus, the broad goal of the current research is to examine infants' visual experiences of objects when infants can move their heads and bodies and interact with objects in a 3D environment. To do so, we employ mobile eye trackers (METs, Fig 1), which have been developed to investigate mobile infants' visual experiences in more naturalistic settings [15, 17, 21]. Two cameras on a headband simultaneously record infants' eye movements and their approximate head-centered field of view, revealing how movements of body in space, and the head within the body, shape what is in view from moment to moment. Most prior MET work asked how

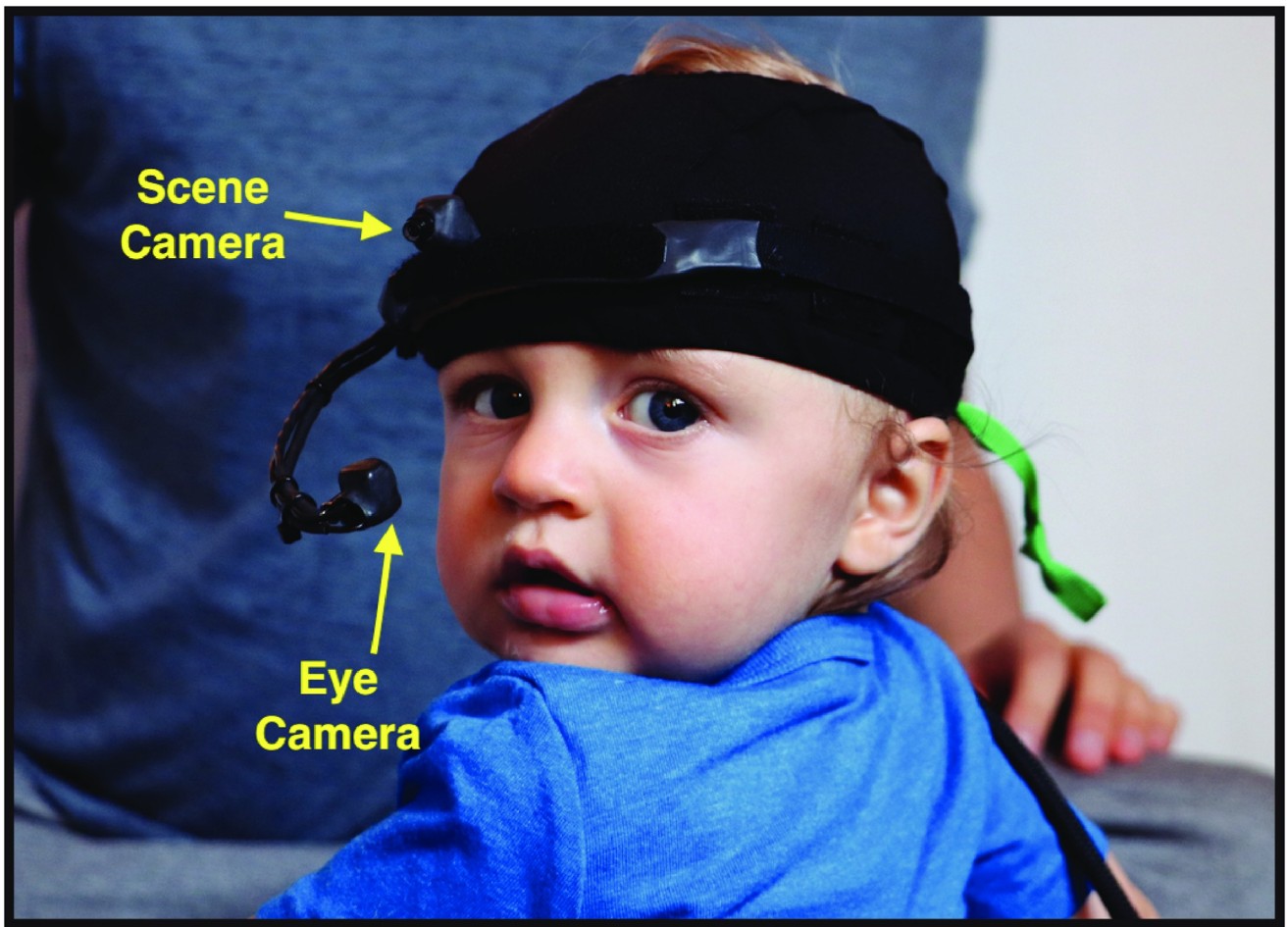

**Fig 1. Mobile eye tracker (MET).** The MET contains a scene camera and an eye camera. The scene camera points straight ahead to record the infant's approximate head-centered field of view. The eye camera points in to the infant's eye to record eye movements.

the motor system influences visual experiences by comparing rates of face/object looking between postures [21] or between observers of different sizes [26]. Little is known about the effect of head movements on visual attention within different postures.

Prior work with adults shows the importance of not only measuring the frequency of gazing at targets but also measuring how the head *centers* targets in view while looking. Previous adult studies show that head movements function to localize gaze targets and stabilize the field of view, which then allows eye movements to make fine-grained movements [27–29]. By using the head to center objects in view, extreme eye movements can be avoided, which facilitates visual processing [30] and makes visual exploration more efficient [31]. For example, inhibiting head movement—and thus removing the ability to center objects in view—decreases the speed and accuracy of object identification [32]. Despite the importance of using the head to center objects in view, little is known about how infants coordinate head and eye movements to orient their visual attention.

Recent work with infants has begun to look at head orientation within a sitting posture to determine how it changes over development [33]. In the study, 9- to 24-month-olds sat in a chair and played with toys at a table while wearing a MET. Infants' first-person field of view was measured using the MET's scene camera, which was worn on infants' foreheads and faced out to capture their view (Fig 1). The locations of toys in the head-centered field of view were measured to reveal how infants' head movements shaped visual experiences of objects. The findings showed that infants moved their head to keep toys centered in view, and indicated that the ability to center information in view improved with age. Furthermore, toys were more centered in view when infants looked at toys versus when they did not look at toys, and centered toys were fixated longer than non-centered toys. This suggests that centering objects in view contributes to the ability to sustain attention, which is a fundamental component of cognitive development [1, 34–36].

However, that study only looked at head movements *within* a single posture, sitting, while the infants were stationary. In everyday life, infants are free to move and *switch* postures (e.g., prone, sitting, upright). How do mobile infants coordinate head and eyes to structure their field of view in different postures? Body posture imposes constraints on infants' ability to access visual information [21, 23–25]. Specifically, infants' view of their surroundings is constrained when prone on all fours compared to when sitting or upright (i.e., standing on two feet). How do infants use the head to center objects in view in different postures that place different motor constraints on looking? These questions remain unanswered.

Another factor that is neglected in most prior work is how the spatial configuration of targets in naturalistic settings impacts visual experiences. In one notable exception, Yamamoto and colleagues [37] showed that the distance between infant and caregiver affects infant-caregiver mutual gaze: Infants and caregivers are less likely to make eye contact if they are too close to one another or too far apart. Similarly, object distance may affect infants' visual attention to objects, but this has yet to be tested since most work employs SETs. Although experimenters can display images of objects on a screen that convey distance information through pictorial cues, distance information in an image lacks behavioral relevance—infants cannot reach to an apparently close object nor to an apparently far object on a computer monitor. And in past work that did use METs during table-top play with real toys, toys were always near infants [18, 19, 33]. Because distance could not vary, it is not clear whether infants can center close and far toys equally well. First, the head and body movements needed to center close and far toys might differ, and may differ within postures. Far toys are located higher in infants' field of view while close toys are in relatively lower locations in the view (Fig 2A). Thus, centering of close and far toys may depend on how infants choose to orient their heads in different body

# A. Sitting infant's field of view

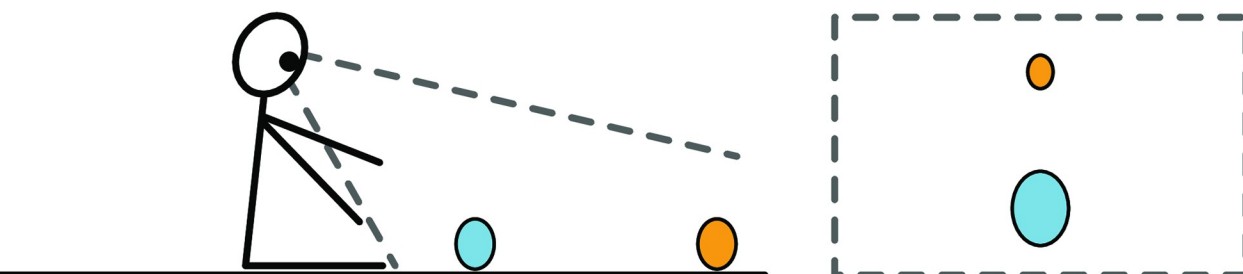

# B. Prone infant's field of view

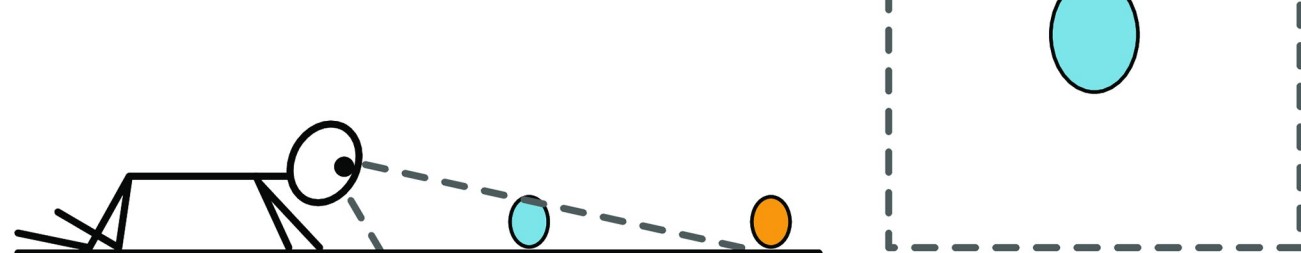

**Fig 2. Schematic drawings of infants' field of view (FOV) in sitting and prone positions.** An infant (A) sitting or (B) prone on the ground with a close toy (blue oval) and a far toy (orange oval). The rectangles on the right of the figure depict infants' approximate first-person FOV in sitting and prone positions (not drawn to scale). (A) In the sitting infant's FOV, close toys are in relatively lower locations, whereas far toys are in relatively higher locations. (B) In the prone infant's FOV, close toys appear large, whereas far toys are out of view.

postures, which, in turn, depends on biomechanical constraints on head movements in each posture. Second, close and far toys have different relevance for motor action. Close toys can be reached to and manipulated whereas far toys cannot be immediately acted on. It is possible that infants tend to look at and center close toys in view for manual exploration, but it may be less important to center far toys in view.

Finally, prior work that studied infants' visual experiences focused only on the frequency of looking—how often infants' eye gaze was fixated on toys during a period of time. However, measuring only looking frequency neglects differences in how visual experiences are structured—what does the field of view contain and in what spatial configuration—and how that structure changes over time. The structure and complexity of object locations in the field of view in real time are important to know because they may affect infant visual attention, and ultimately, learning. Some prior studies suggested that infants' selective field of view may relate to infants' ability to learn novel words [19, 38–41]. Results showed that infants' field of view is often occupied with only a few objects. With fewer objects in view, infants have less difficulty of matching objects with the words they hear, which facilitates word learning. Thus, the structure and complexity of infants' visual experiences—not just the frequency of looking—should be studied to understand how infants' visual experiences may be relevant to learning.

## Current study

The purpose of the study was to investigate how infant posture, toy looking, and toy distance affect the spatial distribution of toys in infants' field of view. This project presents a secondary analysis of [21], in which 12-month-olds played with caregivers in a room with six toys. Infants could freely move around (e.g. crawl, cruise, walk) or remain stationary (e.g., sit, stand), which allowed examining infants' visual experiences in different postures in a naturalistic context. We focused on toys rather than faces in this study for three reasons: 1) the distance of toys changes their relevance to behavior (e.g., close toys can be manipulated, far toys can not), 2) centering objects facilitates sustained attention [33], which is an important component of learning words for novel objects [41–44], and 3) infants rarely look at faces but frequently look at toys in naturalistic play [15, 17, 18, 21], yielding sufficient data to study toy distributions but not face distributions.

The first aim of the study was to test how the relative position between infants and toys (i.e. infant posture and toy distance from infant) may affect infants' visual experiences. Like [33], the current study measured the centering and spread of toys in infants' field of view as they reflect the role of infants' head orientation in structuring visual experiences. Infants wore a MET which recorded their first-person field of view and eye movements (Fig 1). Human coders identified the location of toys in each frame of the field of view videos to determine the distance of each toy to the center of the field of view. The mean and standard deviation of each toy's distance to the center were defined as *centering* and *spread*, respectively. We hypothesized that the centering and spread of close/far toys in infants' field of view would vary across postures. Centering and spread of visual information in the field of view may relate to sustained attention [33] and visual selection [19], which are pivotal to infant learning. By measuring the centering and spread of close and far toys in the field of view, we will determine how infants' visual experiences of objects at different distances are shaped by head orientation in different postures.

The second aim of the study was to test whether infants, as active explorers, can compensate for the constraints imposed by posture and toy distance to center targets of interest in view when looking at toys. In the current study, eye movement data captured by the MET determined *looking episodes*—when infants foveated each toy by pointing the eyes directly at the toy. As visual experiences are shaped by where infants choose to look, we hypothesized that infants' decisions to look at a toy would alter toy locations in the field of view. Past work showed that toys become more centered and less spread in the field of view during looking episodes compared with non-looking episodes [33]. However, the extent to which infants' looking can modify their view of toys may depend on posture and toy distance. For instance, it is more difficult to lift the head to look at far toys in prone position, which may constrain whether infants are able to bring far toys to the center of the view while prone even when they choose to look at toys.

It is reasonable to ask why we would focus on infants' visual experiences of objects at moments that the object is not fixated. First, it should be pointed out that what appears in infants' field of view (even though infants are not looking) is not random. Infants' head and body movements shape the content and properties of the field of view, even if these changes are unintentional byproducts. For instance, infants' field of view is substantially different while crawling versus walking: Crawling infants have much less access to distant and elevated objects compared to walking infants [25]. Those differences in the field of view while infants are not looking reflect the impact of infants' head and body movements on their visual experiences. Second, the content and properties of infants' field of view are important because they affect the opportunities infants have to look at a particular stimulus. Just as wearable audio recorders

(e.g., LENA devices) measure the number of words infants are exposed to without regard for whether infants are attending speech [45–48], measuring the properties of objects in the field of view while infants may not be looking provides insights into the variability in what infants are visually exposed to. Thus, the degree to which infants' head and body movements structure the content of the field of view reflects the processes that shape infants' opportunities for visual learning about objects.

## Method

This study used a previously published MET video dataset [21] to score new behaviors to address the questions above. The data [49], including procedural videos and coding materials, were downloaded from Databrary (Databrary.org), an online digital data library that stores and shares videos for developmental researchers. First, we briefly describe aspects of the original study relevant to the current investigation. Afterwards, we explain the new coding and analyses conducted in the current study, which are shared on OSF (DOI: 10.17605/OSF.IO/3M8JB).

### Description of original study

**Ethics statement.** All experimental protocols and consent materials were approved by the New York University Institutional Review Board. Parents of all participating infants provided written informed consent. The legal guardians of the individuals whose pictures appear in this manuscript have given written informed consent (as outlined in PLOS consent form) to publish these case details.

**Participants.** The final sample of the original study included 17 caregiver-infant dyads recruited from the New York City metropolitan area. Most participants were White and middle class. The age range of the infants was 11.8-12.4 months (*M* age = 12.10, *SD* = 0.18). Eight infants (5 male, 3 female) could sit, crawl, and stand but had not yet begun to walk; of the 8 non-walkers, 7 could walk with external support (e.g. cruising on furniture or with support from their caregiver). The remaining 9 infants (4 male, 5 female) could sit, crawl, stand, and walk both independently and with support. Thus, all infants in the sample were capable of being in sitting, prone (i.e., crawling), and upright (i.e., standing, supported walking, and independent walking) positions.

**Apparatus.** Infants wore a Positive Science head-mounted eye tracker attached to a hat. The eye tracker has two cameras (Fig 1): a *scene camera* and an *eye camera*. The scene camera was positioned above infants' right eye and faced out to capture infants' first-person field of view (54.4˚ × 42.2˚ field of view). Given that the distance between the right eye and head midline is very small in infants, the field of view captured by the scene can be treated as an approximate head-centered field of view. The *eye camera* pointed to the infants' right eye to record eye movements. The field of view videos were 640 pixels horizontal by 480 pixels vertical recorded at 30 frames/sec. Infants also wore a harness which was held by an experimenter who followed infants throughout sessions to keep infants from falling.

The experiment took place in a laboratory space (4.5 m × 6 m). Six common toys (jingling apple, xylophone, ball, plush dog, saxophone, and plastic car) were distributed all over the room for infants to play with during the session. An overhead camera recorded the whole room and a hand-held digital camcorder operated by an assistant captured a closer third-person view of the participants.

**Procedure.** The experimenter put on the eye tracker for infants. To calibrate the eye tracker, the experiment displayed a gridded board about 1 m in front of infants and rattled a small toy to draw infants' attention to specific locations on the board [15, 21, 25]. The

experimenter adjusted the position of the scene camera so that the entire calibration board was visible in the scene video [25] to calibrate the scene camera, approximating a head-centered field of view. Although the angle of the scene camera could not be perfectly equated across infants, we expect the differences between infants would be small because the scene camera has a limited range of motion and the scene camera must be placed in a consistent location above the eye due to the configuration of the eye-tracking headgear. The experimenter also ensured the eye camera captured the right eye's pupil and corneal reflection across the full range of eye movements. By measuring the position of the eye while looking at different locations on the calibration board, the eye tracker calculated gaze location. After the calibration, the play session started. Infants and caregivers played with six toys in a room for 5-15 minutes. Caregivers were told to play with their children as they usually did at home. Both caregivers and children could move around freely in the room. The same procedure of calibration was conducted again at the end of the session to ensure eye tracking accuracy and consistency. The spatial accuracy of gaze location estimates for the eye-tracking data was $M = 1.55°$ of error [21].

**Infant posture coding.**   Infant posture was coded in the original study for each frame as *prone*, *sitting*, or *upright*. Infants in prone posture could be on their hands and knees or lying on their belly. Infants in sitting posture could be tripod sitting, independent sitting, or kneeling with legs tucked under the bottom. Infants in upright posture could be standing or walking with/without support. The moments when infants were in other postures, such as being held by caregivers, were rare and excluded from analyses (<1% of the data).

## Secondary data coding and processing in current study

In the current study, we coded the videos to determine: 1) toy locations in infants' field of view, 2) looking episodes of each toy, and 3) toy distance from infants. Two outcome measures, centering and spread, were calculated based on toy locations to describe infants' visual experiences.

**Toy location in infants' field of view.**   Human coders went through the first 5 min of infants' field of view videos frame by frame and identified any toy that appeared in the view by drawing regions of interest (ROIs) around them using the Dynamic ROI Coder for Matlab v0.2 (www.github.com/JohnFranchak/roi_coding) (Fig 3). The locations of ROIs in the screen were measured in pixel coordinates. The X and Y coordinates of each ROI's center represented *toy location* in infants' field of view. Despite the relatively brief play session, the total dataset across the 17 infants comprised 323,217 hand-coded video frames.

Because of the large volume of data and the slow, manual coding procedure, 8% of each infant's data was randomly selected to be coded by a second independent coder to evaluate inter-rater reliability of ROIs. The proportion of frames on which the coders agreed whether a toy was in view was close to ceiling (*M* across toys = .99). The average correlation between coders for six toys' X coordinates was $r = .97$ and $r = .93$ for Y coordinates. The average location differences between coders were 8.44 pixels for X coordinates and 6.84 pixels for Y coordinates. Thus, coders were able to identify the presence and location of toys with excellent reliability.

**Looking episodes.**   Looking episodes for each toy were defined as frames during which infants looked at that toy. Eye movement data determined the gaze location (green cross in Fig 4) in infants' field of view in pixel coordinates. It was previously reported that the spatial accuracy of gaze location for these gaze data was $M = 1.55°$ of error [21]. To account for the uncertainty in gaze position, ROIs were enlarged to the bounding ellipse (yellow ellipse in Fig 4) to avoid underestimating looking episodes. If eye gaze fell within or on the edge of the ellipse, the

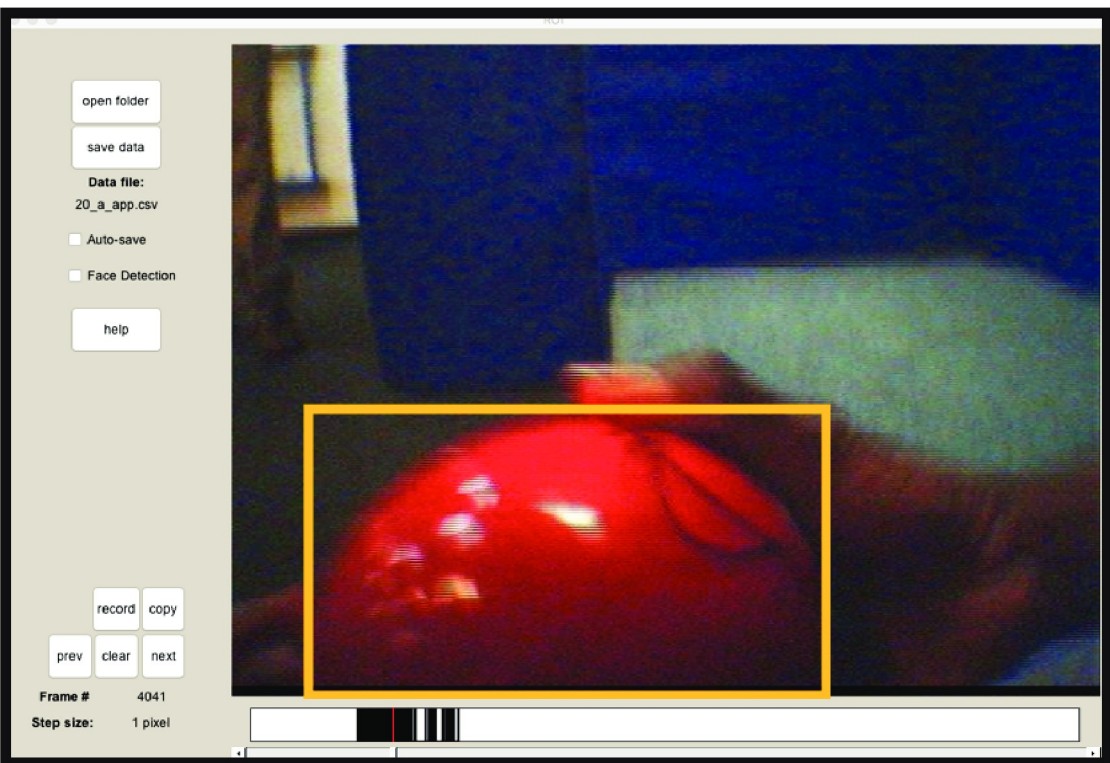

**Fig 3. Dynamic ROI coder.** Coders used the Matlab program *Dynamic ROI Coder* to draw a box around the toy in each frame of infants' field of view video.

frame was defined as a *looking episode* of the toy. Otherwise, it was a *non-looking episode* of that toy. Because it was possible for elliptical ROIs to overlap, infants could possibly be counted as looking at more than one toy in a given video frame. On average, $M = 29.57\%$ ($SD = 14.22$) of video frames were defined as looking episodes, and $M = 70.43\%$ ($SD = 14.22$) were non-looking episodes in the data analyses.

**Toy distance.** Toy distance from infants was coded using Datavyu (Datavyu.org). Coders went through third-person view videos and scored the distance of each toy from the infant for every frame. Toy distance was scored dichotomously as *close* or *far* according to a reference distance: the distance between the infant and the experimenter who held the falling protection harness for the infant (Fig 5). This reference distance was chosen because as the infants moved around in the room, the experimenter kept a relatively consistent distance (∼0.5 m) with the infants while holding the harness. Having the experimenter as a fixed reference point allowed coding of distance as the infant/experimenter moved in depth within the fixed camera used for coding. If a toy's distance to the infant was greater than the reference distance from the infant, the toy was counted as far. Otherwise, the toy was scored as close. All frames in all videos (100%) were coded by two independent coders. Inter-rater agreement for distance codes was reached $M = 97.02\%$.

**Centering and spread.** On every video frame, we calculated the distance between the each toy's location and the center of the infant's field of view (red line in Fig 4). Then, for each infant, we aggregated all toys that were in view for every combination of independent variable levels (infant posture, toy looking, and toy distance), and calculated two outcome variables. The first outcome variable was *centering*: the mean distance of a toy to the center of the field of

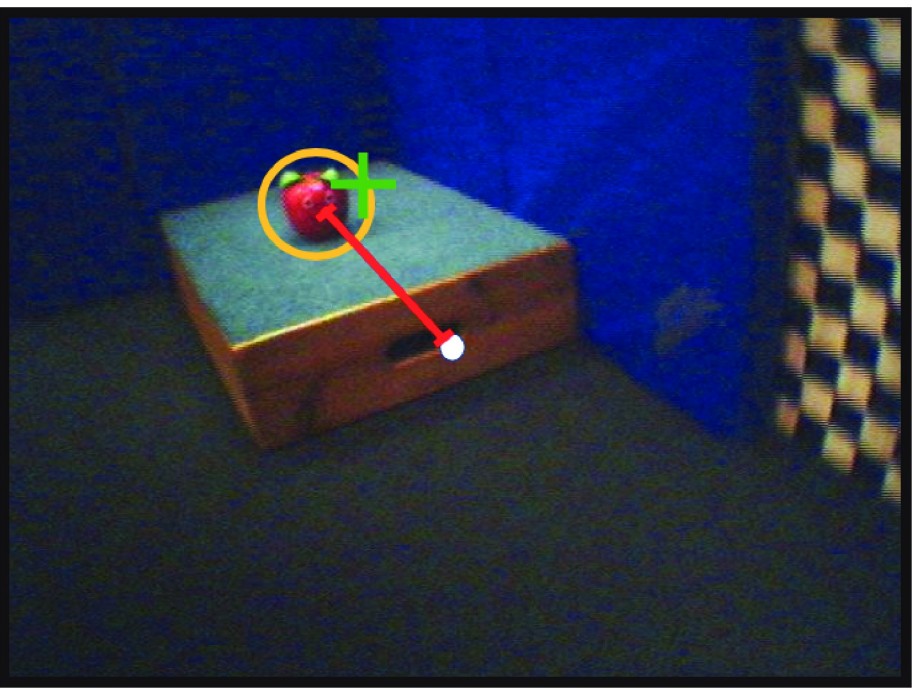

**Fig 4. Looking episode and centering.** A frame was defined as a *looking episode* when eye gaze (green cross) fell on or within ROI (yellow ellipse). Centering (red line) was the distance of toy location to the center of the field of view (white dot).

view. The second outcome variable was *spread*: the SD of each toy's distance to the center of the field of view. Thus, centering reflected the average centering of toys in the field of view and spread described the variability of centering for a given combination of conditions (e.g., looking at close toys while prone).

**Statistical approach.** Linear mixed models (LMMs) were applied to test the effects of posture, looking, and toy distance on both centering and spread. For each outcome variable, a separate LMM was calculated using the *lme4* [50] package in R [51] with posture, looking, and toy

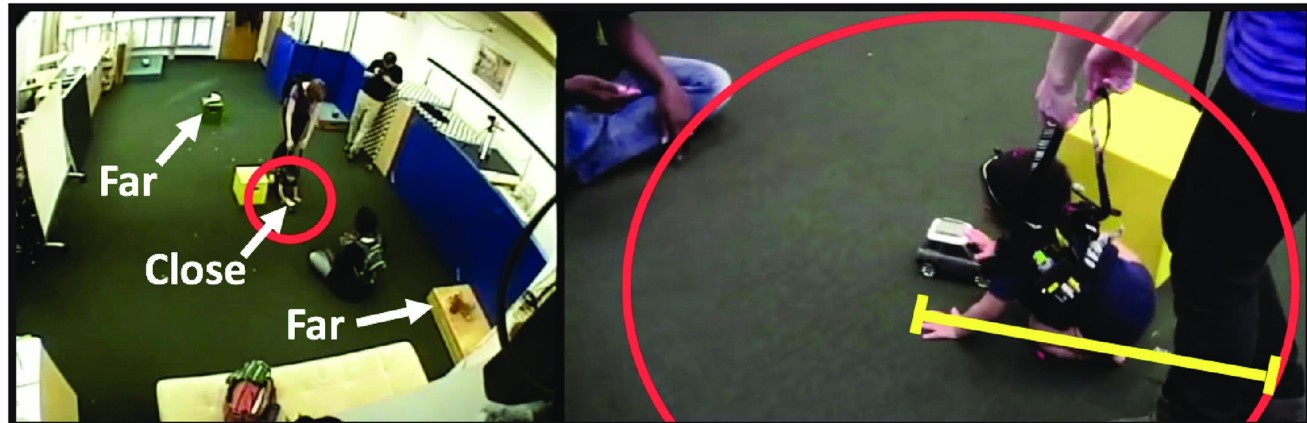

**Fig 5. Distance coding.** Coders went through third-person view videos to score toy distance. Toys within the reference distance (yellow line) were coded as close to the infant, whereas toys beyond the reference distance were coded as far.

distance (and their interactions) as fixed factors and with random intercepts for participant (random slope models failed to converge). Posture, looking, and toy distance were treated as categorical factors with three levels of posture (prone, sitting, and upright), two levels of looking (looking, non-looking), and two levels of toy distance (close, far). ANOVAs were used to test significance of main effects and interactions using the *lmerTest* package in R [52]. Degrees of freedom were determined by the Satterthwaite approximation [53, 54]. Follow-up pairwise comparisons used the Holm-Bonferroni correction to adjust for multiple comparisons.

## Results

### Centering of toy locations in infants' field of view

As Fig 6A shows, posture and distance moderated the degree to which toys were centered in infants' field of view. While prone, far toys were more poorly centered in the field of view compared to when sitting and upright. In addition, toys were more centered in the field of view during looking episodes compared to non-looking episodes regardless of posture and toy distance. Table 1 lists descriptive statistics for centering according to posture, looking, and toy distance.

A LMM predicting centering from posture, looking, and toy distance confirmed the main effects of posture and looking and a significant posture × toy distance interaction (Table 2). To follow up on the significant posture × toy distance interaction, pairwise comparisons were calculated separately for far toys and close toys by posture while collapsing across looking and non-looking episodes. Far toys were less centered while prone compared to sitting ($t(164) = 4.20$, $p < .001$) and upright ($t(166) = 4.14$, $p < .001$). Centering for far toys did not differ between sitting and upright ($t(162) = .08$, $p = .935$). In contrast, the centering of close toys was similar across postures ($p$s $> .05$). Thus, posture influenced infants' visual experiences of far toys but not close toys.

### Spread of toy locations in infants' field of view

The spread of toy locations in infants' field of view was related to posture, looking, and toy distance (Fig 6B). Overall, the spread was smaller during looking episodes compared to non-looking episodes, suggesting toy locations were more concentrated in the field of view at the moment of looking. During non-looking episodes (Fig 6B left panel) specifically, toy locations were more spread out in sitting and upright positions compared to prone. Far toys' locations were more variable than close toys' locations. In contrast, during looking episodes (Fig 6B right panel), close toys were more spread out than far toys in prone position, whereas the spread of close and far toys were similar in sitting and upright positions. Table 3 shows the means and standard deviations of spread for each combination of posture, looking, and toy distance.

A LMM predicting spread from posture, looking, and toy distance confirmed a significant looking effect, a significant looking × toy distance interaction, and a significant posture × toy distance interaction (Table 4). Overall, the spread of toy locations decreased by 21.17 pixels from non-looking to looking episodes ($M_{non-look} = 73.66$, $M_{look} = 52.49$, $t(162) = 10.68$, $p < .001$). To follow up on the looking × toy distance and posture × toy distance interactions, a LMM predicting the spread from posture and toy distance was conducted separately for looking episodes and for non-looking episodes.

Within non-looking episodes, the results of LMM confirmed the main effects of posture ($F(2, 77.54) = 3.23$, $p = .045$) and toy distance ($F(1, 76.80) = 14.45$, $p < .001$). Visual inspection of Fig 6B (left panel) suggested that the main effect of posture (across toy distance) is due to increasing spread from prone to sitting and from sitting to upright. However, pairwise

## A. Centering

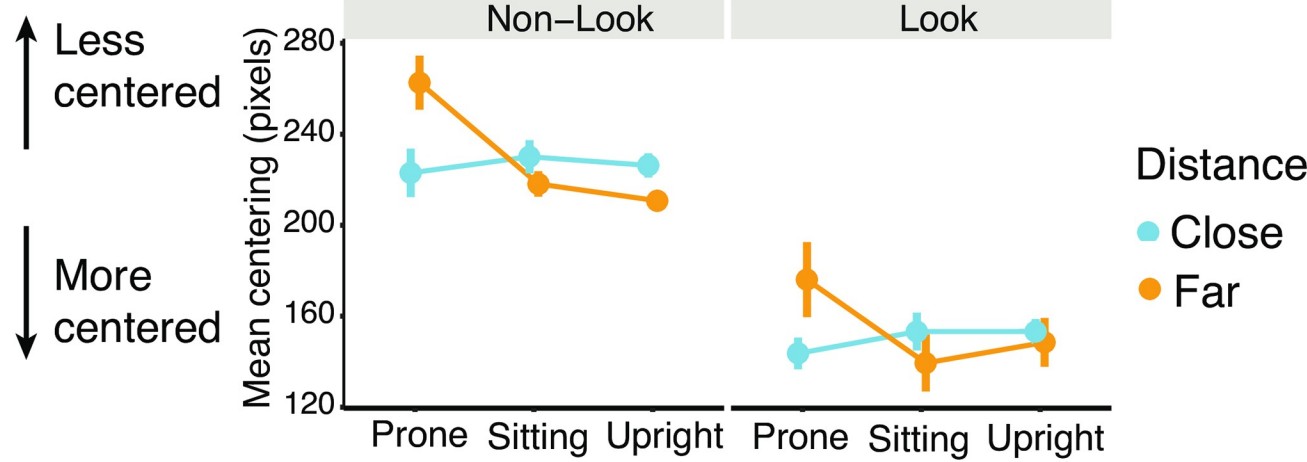

## B. Spread

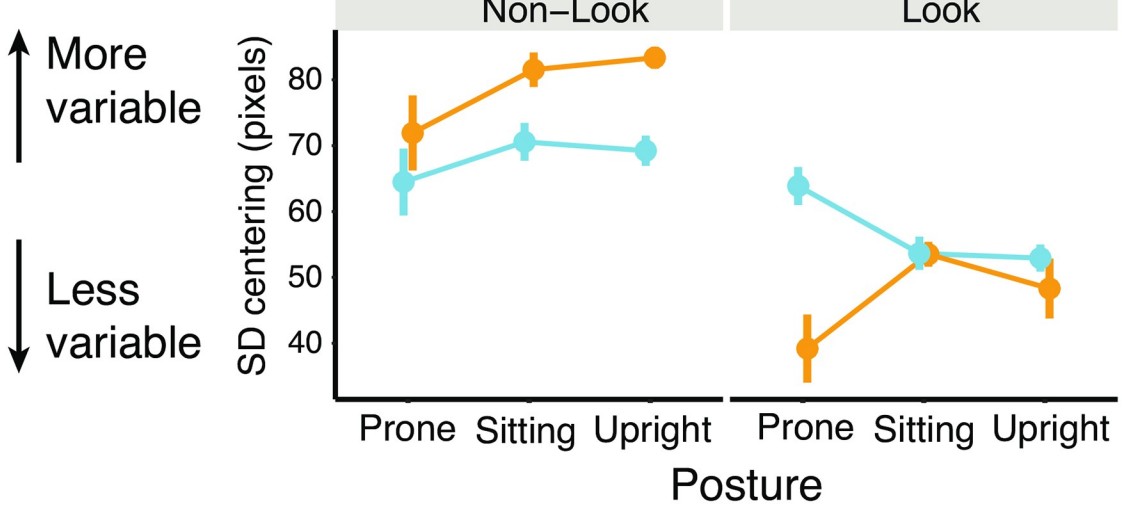

**Fig 6. Centering and spread by posture, toy looking, and toy distance.** (A) Centering and (B) spread of close toys (blue symbols) and far toys (orange symbols) during looking and non-looking episodes across postures. For centering, smaller values on the y-axis indicate better centering. For spread, smaller values on the y-axis indicate less variability of toy locations in the field of view.

comparisons between postures were not statistically significant. The main effect of toy distance indicated far toys had larger spread than close toys ($p < .001$).

Posture and toy distance influenced spread differently for looking episodes. The results of LMM for looking episodes showed a main effect of toy distance ($F(1,85) = 11.76$, $p < .001$), and a posture × toy distance interaction ($F(2,85) = 6.43$, $p = .003$). Pairwise comparisons followed up the posture × toy distance interaction. Results suggested that during looking episodes, the spread was larger for close toys than far toys in prone position ($t(73.3) = 4.59$, $p < .001$). But there was no difference between close and far toys in sitting ($t(71) = .03$, $p = .977$) and upright ($t(70.2) = .98$, $p = .330$).

**Table 1. Descriptive statistics (*M* and *SD* in pixels) for centering by posture, looking, and toy distance.**

| | Non-looking | Looking |
|---|---|---|
| **Prone** | | |
| Close | 223.01 (43.89) | 143.62 (28.70) |
| Far | 262.68 (49.00) | 176.10 (68.01) |
| **Sitting** | | |
| Close | 230.11 (29.99) | 153.25 (34.21) |
| Far | 218.15 (22.96) | 139.43 (51.59) |
| **Upright** | | |
| Close | 226.31 (21.89) | 153.28 (22.59) |
| Far | 210.78 (18.28) | 148.48 (44.15) |

**Table 2. Summary of LMM results predicting centering from posture, looking, and toy distance.** Degrees of freedom and resulting *p* values obtained using the Satterthwaite approximation.

| | df | F | p | |
|---|---|---|---|---|
| **Posture** | 2, 163.27 | 3.86 | .023 | * |
| **Looking** | 1, 161.16 | 207.04 | <.001 | *** |
| **Distance** | 1, 161.53 | .75 | .387 | n.s. |
| **Posture×Looking** | 2, 160.99 | .68 | .508 | n.s. |
| **Posture×Distance** | 2, 161.41 | 8.77 | <.001 | *** |
| **Looking×Distance** | 1, 161.22 | .006 | .941 | n.s. |
| **Posture×Looking×Distance** | 2, 160.97 | 0.23 | .795 | n.s. |

* *p* < .05,

** *p* < .01,

*** *p* < .001.

**Table 3. Descriptive statistics (*M* and *SD* in pixels) for spread by posture, looking, and toy distance.**

| | Non-looking | Looking |
|---|---|---|
| **Prone** | | |
| Close | 64.48 (20.90) | 63.88 (11.86) |
| Far | 71.93 (23.45) | 39.20 (21.30) |
| **Sitting** | | |
| Close | 70.57 (11.82) | 53.65 (10.39) |
| Far | 81.53 (10.76) | 53.52 (7.75) |
| **Upright** | | |
| Close | 69.23 (9.45) | 52.92 (8.45) |
| Far | 83.34 (6.86) | 48.31 (18.74) |

## Discussion

This study investigated how infants' visual experiences of toys during play depend on the physical relation between infant and toy, which is created by infants' eye, head, and body movements. Specifically, we found that the centering and spread of toy locations in infants' field of view were affected by infant posture, looking, and toy distance. Overall, infants' decision to look at toys led to better centering and decreased spread of toy locations, regardless of toy

**Table 4. Summary of LMM results predicting spread from posture, looking, and toy distance.** Degrees of freedom and resulting p values obtained using the Satterthwaite approximation.

| | df | F | p | |
|---|---|---|---|---|
| Posture | 2, 163.35 | 1.97 | .143 | n.s. |
| Looking | 1, 160.91 | 114.24 | <.001 | *** |
| Distance | 1, 161.37 | .05 | .825 | n.s. |
| Posture×Looking | 2, 160.69 | 1.50 | .226 | n.s. |
| Posture×Distance | 2, 161.21 | 4.78 | .010 | ** |
| Looking×Distance | 1, 160.98 | 26.32 | <.001 | *** |
| Posture×Looking×Distance | 2, 160.66 | 2.23 | .111 | n.s. |

\* $p < .05$,

\*\* $p < .01$,

\*\*\* $p < .001$.

distance and posture, demonstrating how attention serves to structure visual experiences amidst variations in the spatial relation between infant and object.

Beyond the overall effect of looking, complex interactions between posture, distance, and looking show the value of studying attention in mobile infants in a naturalistic play task as opposed to in a stationary screen task. First, far toys were less centered in the field of view while infants were prone compared to when they were sitting and upright. However, posture did not influence the centering of close toys. It implies that infants while prone have a disadvantaged access to far toys but not close toys. This moderating effect of distance may be due to infants while prone pointing their head down—restricting how far the upper bound of the field of view extends in the distance [25]. Consequently, far toys are usually located near the top edge of the field of view. Second, toy locations in the field of view became more spread from prone to sitting and from sitting to upright during non-looking episodes, regardless of toy distance. Most likely, this results from a difference in elevation of the head—and thus viewpoint—between upright, sitting, and prone positions. As the head is elevated higher off the ground, the field of view encompasses a larger area of the environment which allows for a larger spatial distribution of object locations in the field of view. In contrast, during looking episodes the spread of toys in the field of view varied only between close and far toys while prone. This shows that through the act of looking, infants focus on a single object and actively maintain it in the center of view through head movements, resulting in a narrow spread. The reason why the spread of close toys while prone is larger compared to in sitting and upright positions during looking episodes may be because infants have more difficulty stabilizing the field of view while prone. Tilting the head up and down is effortful while prone, which may hinder head movements to re-center objects in view. Thus, infants are less likely to keep the toys at consistent locations in the field of view when prone, but do so more easily when the range of head movement while sitting and upright is relatively unrestricted.

Documenting visual experiences and how they are shaped by gross motor behavior is important because infants learn from what they see. Most previous studies investigated infants' visual experiences by reporting only the frequency of looking at different targets, such as objects and faces [18, 21]. Although looking frequency tells us what opportunities infants have to learn about different categories of visual stimuli in different situations, it overlooks the spatial aspect of infants' visual experiences—where and how objects are distributed in the field of view at each moment. The spatial aspects of visual experiences are important in that infants select what to attend in the field of view in real time *actively* through eye, head, and body

movements. How centered and complex the distribution is affects infants' visual attention and, potentially, learning. Studies with adults suggest that centering objects in the field of view by aligning eyes and head facilitates visual processing [31, 55]. Moreover, past developmental work indicates that infants' field of view is typically dominated by one object, which decreases the difficulty of visual selection for infants and facilitates learning [19, 39]. Another study showed that infants tend to center visual information in the field of view, which may support sustained attention [33]. However, those studies were based on settings where infants were sitting at a table and toys were placed very close to them.

But infants are mobile in everyday life, and the current study goes a step farther than past work to reveal how infants shape visual experiences in the context of mobility. This study is novel in investigating the object distribution in infants' field of view when infants can freely move their body in different postures and move closer and farther from objects. Our results extend past work showing that infants center objects in their field of view when looking [33] by showing that they do so in prone and upright postures, not just sitting, and they do so for far objects, not just close objects. Also, the current study found that infants are overall exposed to increasingly variable visual experiences from prone to sitting and from sitting to upright (among non-looking episodes, which comprised most visual experiences of objects). This implies the acquisition of new motor abilities, such as during the transition from crawling to walking, may increase the complexity of infants' visual experiences. Whether this helps or hinders learning has yet to be tested. Complex and variable visual experiences may further encourage infants to move and explore the environment, promoting both motor and cognitive development (e.g. spatial memory) [56, 57]. Alternatively, the more complex and cluttered experiences in some postures, such as when upright, could be a hindrance to learning. For example, pairing object labels to the correct visual referent may be more difficult when upright compared to when sitting if upright infants experience a more cluttered visual scene. The downstream effects of changing visual experiences in development should be studied in the future to discern between these possibilities [58].

Differences in visual experiences in different postures are amplified when object distance is taken into account. The current study differentiated close and far objects in the environment—another benefit to testing visual experiences in a mobile task as opposed to a seated task. We found that infants while prone have difficulty centering far objects in view compared to when sitting and upright, perhaps because lifting the head up to look at far objects is effortful while prone. As a result, far objects are more likely to reside at the top of their field of view and be less centered. But in spite of the constraints imposed by posture and environment, infants are by no means passive observers. In contrast, they actively structure their visual experiences by coordinating head and eyes to look. Regardless of the differences across postures, within every posture infants actively centered and decreased the spread of toy locations when choosing to look at toys. Although infants may not be able to overcome the constraints of a posture completely, they do compensate for the postural and environmental constraints to some degree when they choose to look, using the body to facilitate sustained object attention.

We acknowledge several limitations in the study. First, we only considered whether infants' posture affected visual experiences, but not whether infants were stationary or mobile. When upright, infants can be standing still, cruising along furniture, or walking independently from one place to another. Likewise, while prone infants could have been either stationary or crawling. Future work should compare infants' head orientation when stationary versus moving. It is possible that infants look more at close toys when stationary while more at far toys when locomoting. The biomechanics of maintaining balance and need for visual guidance while moving versus stationary may also impact how infants orient the head. A second variable that was not considered was the elevation of toys off the ground. Elevation, not just distance,

determines how the head must be oriented to center a toy in view. A previous study suggested that infants in different postures have different access when toy height was manipulated [25]. Infants while prone have more difficulty lifting their head to look at objects at high elevations, especially when toys are close. In the current study, toys could be placed on the ground, but could also be picked up by infants and caregivers and placed on low platforms in the room. Future studies should test how toy elevation may affect infants' centering of information in the field of view. Last, this study did not investigate the relation between the spatial distribution of objects and infants' hand action. It is unknown whether the properties of visual experiences affect the probability of infants' reaching toward objects, and how visual experiences change as visual system guides reaching. Future work should look at the association between infants' visual experiences and hand actions.

In conclusion, the findings suggest that infants have different visual experiences of close and far objects in different postures. Infants are constrained by the body and environment. Head orientation is constrained by the body posture: Tilting the head up and down is more effortful while prone, resulting in distinct visual experiences. Also, infants' visual experiences are affected by their spatial relationship with the objects in the environment—whether toys are located close versus far from infants. Nevertheless, infants actively coordinate the head and eyes to choose what in view and structure the location of visual information in view. The resulting spatial characteristics in the field of view may affect infants' visual attention in real time and learning over developmental time.

## Acknowledgments

We thank Karen Adolph and Kari Kretch, co-authors of the original study, for sharing this dataset on Databrary to allow reanalysis. We also thank the members of UCR Perception, Action, and Development Lab for their assistance in coding the data.

## Author Contributions

**Conceptualization:** Chuan Luo, John M. Franchak.

**Data curation:** Chuan Luo, John M. Franchak.

**Formal analysis:** Chuan Luo, John M. Franchak.

**Methodology:** Chuan Luo, John M. Franchak.

**Project administration:** Chuan Luo.

**Resources:** John M. Franchak.

**Software:** John M. Franchak.

**Supervision:** John M. Franchak.

**Visualization:** Chuan Luo, John M. Franchak.

**Writing – original draft:** Chuan Luo, John M. Franchak.

**Writing – review & editing:** Chuan Luo, John M. Franchak.

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
