## [Decision Letter · Decision Letter 0]

28 Aug 2020

PONE-D-20-22503

Head and body structure infants’ visual experiences during mobile, naturalistic play

PLOS ONE

Dear Dr. Franchak,

Thank you for submitting your manuscript to PLOS ONE. After careful consideration, we feel that it has merit but does not fully meet PLOS ONE’s publication criteria as it currently stands. Therefore, we invite you to submit a revised version of the manuscript that addresses the points raised during the review process.

Please see my comments, below.

We look forward to receiving your revised manuscript.

Kind regards,

Thomas A Stoffregen, PhD

Academic Editor

PLOS ONE

Journal Requirements:

2. Please provide additional details regarding participant consent. In the ethics statement in the Methods and online submission information, please ensure that you have specified whether consent was informed.

4.We note that Figure [1] includes an image of a patient / participant in the study. 

Additional Editor Comments (if provided):

I have received two reviews from experts in the field. One requests quite substantial changes, while the other suggests relatively minor changes. I have categorized things as "major revision", though I'd prefer a category intermediate between major and minor. In any event, I trust you will carefully consider both reviews, and I hope (and expect) you will be able to submit a suitable revision.

Reviewers' comments:

Reviewer's Responses to Questions

**Comments to the Author**

1. Is the manuscript technically sound, and do the data support the conclusions?

Reviewer #1: Yes

Reviewer #2: Yes

2. Has the statistical analysis been performed appropriately and rigorously? 

Reviewer #1: Yes

Reviewer #2: Yes

3. Have the authors made all data underlying the findings in their manuscript fully available?

Reviewer #1: Yes

Reviewer #2: Yes

4. Is the manuscript presented in an intelligible fashion and written in standard English?

Reviewer #1: Yes

Reviewer #2: Yes

5. Review Comments to the Author

Reviewer #1: This manuscript describes the re-analysis of the eye-tracking data of 12-month-old infants in a mobile play setting. The authors examined how centered and variable toy locations in infants’ field of view, and investigated how infant posture, toy looking, and toy distance affected the centering and variability. Beyond the overall effect of toy looking, systematic interactions were found between posture, distance, and toy looking. Among others, the results showed that far toys were less centered in the field of view while infants were prone compared to when they were sitting and upright, and that close toys were more variable in the field of view compared to far toys while infants were in prone. The authors concluded that infants have different visual experiences of close and far objects in different postures, and that infants actively coordinate the head and eyes to look at an object. For the most part, the study described in this paper was well-motivated and competently conducted. However, I have a couple of concerns about the method and the interpretation of the results that need to be addressed before I could recommend publication.

1. Head-centered field of view

How was the center of the scene camera image aligned to the center of the field of view of the infant’s right eye (i.e., how was the head-centered field of view defined)? I believe that calibration in the eye tracker is done so as to match the position of the pupil in the eye-camera image and the location of the object in the scene camera image. The eye tracker can track the gaze accurately even if the orientation of scene camera is not perfectly aligned to the center of the actual field of view of the right eye. Slight changes in the orientation of the scene camera could result in different “center” in the scene camera image. In other words, although the eye tracker is able to pick up the location of gaze (green cross in Fig. 4) accurately, the center of the scene camera image and resulting distance between the image center and the green cross may not be so accurate.

2. Non-looking episodes

By definition, the toys in non-looking episodes were what randomly got in infant’s field of view. They were not something that infant’s visual system was oriented to. As such, I’m not sure whether and how the analysis of the data of “non-looking” episode could deepen our understanding the nature of infants’ visual experiences during naturalistic tasks. For instance, what does the result imply that far toys are less centered compared to close toys not only in looking episodes but also in “non-looking” episodes when infants are in prone (Fig. 6A)? Why did toy locations in the field of view become more spread from prone to sitting and from sitting to upright during “non-looking” episodes, but not during looking episodes (Fig. 6B)? What underlies the increased spread in the gaze toward closely located toys while infants are prone in looking episodes (Fig 6B), which was the exact opposite to the tendency found in “non-looking” episode just mentioned above? I have difficulty in interpreting the results of the analysis of what infants were not looking at, and the comparison between looking and non-looking episodes.

3. The effect of distance

The authors used the distance in the 2D image of the scene camera to measure how centered and variable toy locations in infants’ field of view. Close toy occupies the greater area of the scene camera image (Fig 3) than far toy does (Fig 4). I wonder if there are greater chances for eye gaze (green cross) to fall near the ROI when a toy is closely located and takes up the scene camera image compared to the situation where the toy is located far. If the actual distance between the object and the infant is available, visual angle instead of distance on 2D image might be a better measure, as it may compensate for such an effect.

Reviewer #2: Abstract

- No comments

Introduction

- The authors state, “Although it is a useful technique to examine some aspects of infants' selective attention, it precludes measuring the motor aspects of attention and how they shape infants' visual experiences of objects in daily life.” Researchers who use SET to examine infants’ visual attention may classify eye movement as “motor movements.” It would be useful for the authors to specify that they are referring to gross motor movements. This would strengthen their argument about the role of body position on visual attention in infancy.

Method

- Under procedure, is there a publication that can be cited to elaborate on the calibration procedure?

Results

- Please include units for Tables 1 and 3.

Discussion

- I think that an additional limitation or perhaps future direction could be pairing visual attention with the probability of reaching toward particular objects. This would offer information about visually guided reaching and prospective control with upper extremity tasks in infancy.

6. PLOS authors have the option to publish the peer review history of their article (what does this mean?). If published, this will include your full peer review and any attached files.

Reviewer #1: No

Reviewer #2: No

---

## [Author Response · Author response to Decision Letter 0]

12 Oct 2020

Please see attached "Response to Reviewers" letter.

---

## [Decision Letter · Decision Letter 1]

26 Oct 2020

Head and body structure infants’ visual experiences during mobile, naturalistic play

PONE-D-20-22503R1

Dear Dr. Franchak,

We’re pleased to inform you that your manuscript has been judged scientifically suitable for publication and will be formally accepted for publication once it meets all outstanding technical requirements.

Kind regards,

Thomas A Stoffregen, PhD

Academic Editor

PLOS ONE

Additional Editor Comments (optional):

Reviewers' comments:

Reviewer's Responses to Questions

**Comments to the Author**

1. If the authors have adequately addressed your comments raised in a previous round of review and you feel that this manuscript is now acceptable for publication, you may indicate that here to bypass the “Comments to the Author” section, enter your conflict of interest statement in the “Confidential to Editor” section, and submit your "Accept" recommendation.

Reviewer #1: All comments have been addressed

Reviewer #2: All comments have been addressed

2. Is the manuscript technically sound, and do the data support the conclusions?

Reviewer #1: Yes

Reviewer #2: (No Response)

3. Has the statistical analysis been performed appropriately and rigorously? 

Reviewer #1: Yes

Reviewer #2: Yes

4. Have the authors made all data underlying the findings in their manuscript fully available?

Reviewer #1: Yes

Reviewer #2: Yes

5. Is the manuscript presented in an intelligible fashion and written in standard English?

Reviewer #1: Yes

Reviewer #2: Yes

6. Review Comments to the Author

Reviewer #1: The authors have addressed my concerns as well as those of the other reviewers. I believe that the manuscript can now be accepted for publication.

Reviewer #2: The authors have satisfactorily responded to the reviewers' concerns. I think that the paper is now ready to be accepted.

7. PLOS authors have the option to publish the peer review history of their article (what does this mean?). If published, this will include your full peer review and any attached files.

Reviewer #1: No

Reviewer #2: No

---

## [Editor Report · Acceptance letter]

30 Oct 2020

PONE-D-20-22503R1 

Head and body structure infants’ visual experiences during mobile, naturalistic play 

Dear Dr. Franchak:

I'm pleased to inform you that your manuscript has been deemed suitable for publication in PLOS ONE. Congratulations! Your manuscript is now with our production department. 

Kind regards, 

on behalf of

Dr. Thomas A Stoffregen 

Academic Editor

PLOS ONE